# Emerging Pathogen Threats in Transfusion Medicine: Improving Safety and Confidence with Pathogen Reduction Technologies

**DOI:** 10.3390/pathogens12070911

**Published:** 2023-07-05

**Authors:** Marcia Cardoso, Izabela Ragan, Lindsay Hartson, Raymond P. Goodrich

**Affiliations:** 1Terumo BCT, Inc., TERUMO Böood and Cell Technologies, Zaventem, 41 1930 Brussels, Belgium; 2Infectious Disease Research Center, Department of Biomedical Science, Colorado State University, Fort Collins, CO 80521, USA; izabela.ragan@colostate.edu; 3Infectious Disease Research Center, Department of Microbiology, Immunology and Pathology, Colorado State University, Fort Collins, CO 80521, USA; lindsay.hartson@colostate.edu (L.H.); ray.goodrich@colostate.edu (R.P.G.)

**Keywords:** transfusion, blood, pathogen reduction, emerging infectious diseases

## Abstract

Emerging infectious disease threats are becoming more frequent due to various social, political, and geographical pressures, including increased human–animal contact, global trade, transportation, and changing climate conditions. Since blood products for transfusion are derived from donated blood from the general population, emerging agents spread by blood contact or the transfusion of blood products are also a potential risk. Blood transfusions are essential in treating patients with anemia, blood loss, and other medical conditions. However, these lifesaving procedures can contribute to infectious disease transmission, particularly to vulnerable populations. New methods have been implemented on a global basis for the prevention of transfusion transmissions via plasma, platelets, and whole blood products. Implementing proactive pathogen reduction methods may reduce the likelihood of disease transmission via blood transfusions, even for newly emerging agents whose transmissibility and susceptibility are still being evaluated as they emerge. In this review, we consider the Mirasol PRT system for blood safety, which is based on a photochemical method involving riboflavin and UV light. We provide examples of how emerging threats, such as Ebola, SARS-CoV-2, hepatitis E, mpox and other agents, have been evaluated in real time regarding effectiveness of this method in reducing the likelihood of disease transmission via transfusions.

## 1. Introduction

Historically, diseases were spread under the conditions of war, trade, and travel. Today, in the age of globalization, the mobility of goods and people is extremely high [1]. Moreover, human population growth has led to an increased urbanization of wild habitats and the over-exploitation of water and fossil fuels, which are culminating in a remarkable increase in land and ocean temperatures since 1981 [2,3,4]. The effects of these changes are seen in the increased number of floods and intense storms, as well as the thawing of permafrost and melting of sea ice, which will continue to push people and animals into more restricted geographical areas [5]. We now see an increased proximity of humans to wild animals, including primates, as well as the presence of insect vectors in temperate regions previously only found in the tropics [3]. All of these propagate the impact of the disease triangle comprising the environment, pathogens, and society [6].

Over the past 40 years, we have seen an increasing emergence and re-emergence of infectious diseases. HIV was identified as the agent of a pivotal species cross-over infection leading to the global AIDS epidemic. Its spread was facilitated by modern human practices and social behaviors, e.g., sexual activity, blood transfusions, and intravenous drug abuse [7,8]. Outbreaks of severe acute respiratory syndrome coronavirus (SARS-CoV) in Southeast Asia, Ebola virus disease in Africa, Middle East respiratory syndrome coronavirus (MERS-CoV) in the Middle East, and Zika virus disease, chikungunya, yellow fever, and dengue in the Americas followed in the ensuing two decades [9,10]. Finally, in 2020, the WHO declared the COVID-19 outbreak caused by SARS-CoV-2 a pandemic, which was responsible for over 6.8 million human deaths globally [11].

Emerging and re-emerging pathogens pose an important risk to transfusion medicine. Along with the classic bloodborne pathogens, HIV, HBV, and HCV, arbovirus transmission through blood transfusion has been increasingly reported in the past 20 years [12,13]. While there have been numerous reports of transfusion-transmitted infectious by West Nile and Dengue viruses, there are other arboviruses tentatively implicated in disease transmissions, such as Zika virus, yellow fever virus, tick-borne encephalitis virus, Japanese encephalitis virus, Powassan virus, St. Louis encephalitis virus, Ross River virus, and Colorado tick fever virus [14].

In the past 40 years, blood transfusion safety measures have been developed and implemented as a reaction to the identification of new infectious threats. Initially, the detection of infections was based on immunological assays. Currently, HBV/HCV/HIV nucleic-acid-based tests or NATs are part of the blood screening algorithm in many parts of the world. These tests are able to detect pathogens at an earlier phase of infections [12]. In accordance with local epidemiology, WNV-NATs have also been integrated in the blood donation screening strategies of some countries, and most recently, NATs to detect Babesia sp. and Zika virus infections have been implemented in the USA, the latter of which was eventually discontinued [12,15,16,17,18]. Testing strategies have been very effective in reducing risks once identified [19]. Yet, the climate/social developments of the past several years make the emergence of novel pathogens unpredictable and highlight the inadequacy of reactive prevention strategies in quickly addressing emergent needs in a timely way.

Pathogen reduction (PR) is a proactive strategy to mitigate the risk of transfusion-transmitted infections (TTI). Available PR methods involve the physicochemical disruption of pathogen structural elements or the photochemical modification of nucleic acids to prevent replication [20,21,22]. The plasma fractionation industry has the longest and most successful experience with PR, starting 30 years ago with the systematic application of methods of pathogen inactivation/removal in the manufacturing process to increase the safety of plasma-derived medicine products to the high levels seen today [23].

As for labile blood components, methods of PR have been gradually implemented in Europe in the last 10 years and lately in the USA, mostly for the treatment of platelet concentrates and plasma for transfusions [16,24]. Commercially available systems utilize different wavelengths of UV light or visible light with or without different photochemical or photodynamic compounds [21]. The Theraflex MB System (MacoPharma) was developed for the treatment of individual plasma for transfusions [22]. It combined methylene blue as the photoreactive compound and visible light and was the first commercialized system. It was followed by the Intercept system (Cerus Corp) for the treatment of platelets and plasma using the chemical substance amotosalen hydrochloric acid (S-59), a synthetic psoralen, and UV-A. The Mirasol PRT System for plasma, platelets, and whole blood utilizes riboflavin or vitamin B2 as the photosensitizer and UV light. Recently, the Theraflex UV Platelet system was developed by MacoPharma and relies only on illumination with UV-C light.

In 2016, the results of an AIMS randomized clinical trial which investigated the effectiveness and feasibility of a riboflavin + UV light treatment of whole blood for transfusions in a highly malaria-endemic country were published. It showed for the first time the effectiveness of the PRT System in reducing the incidence of transfusion-transmitted disease, in this case, malaria [25,26]. Moreover, the componentization of whole blood treated with riboflavin and UV light and the transfusion of processed red cell concentrates have shown promising results in relation to safety and therapeutic effectiveness in a cohort of pediatric patients [27]. Treatments to inactivate pathogens in red cell concentrates are still under development [22,28].

Doubtless, worldwide adoption of such proactive technology will depend on the availability of a technology for treatment of all blood derived components universally and at acceptable cost. Yet, component treatment has proven that the concept of a universal pathogen-inactivated transfusion is possible, as demonstrated by the recent report of patients receiving all blood components treated with riboflavin plus UV technology [24].

In this article, we will describe the most recent outbreaks of viral pathogens (mpox, SARS-CoV-2, Ebola virus, and hepatitis E virus) with the potential to challenge the established reactive protective measures in place in transfusion medicine. Moreover, we will describe how pathogen reduction technology based on riboflavin (vitamin B2) and UV light may contribute to improving the preparedness of the transfusion community for future emergent virus threats to blood supply safety and integrity.

## 2. Pandemic Preparedness and Response for Emerging Pathogen Threats and the Role of PRT

The rapid emergence and spread of SARS-CoV-2 has emphasized the importance of pandemic preparedness for emerging pathogen threats. Global pandemic preparedness involves a range of strategies and tools to detect, respond to, and control the spread of infectious diseases. For example, in early stages of a disease’s spread, convalescent plasma, obtained from recovered patients, has been used as a potential therapeutic until additional therapies can be developed and deployed. However, the use of convalescent plasma carries risks, especially when the routes of transmission of an emerging pathogen is unknown. This is where PRT can be valuable in mitigating the transfusion transmission risk early in a pandemic.

Several strategies and tools are designed to detect disease outbreaks early, prevent and control their spread, and reduce their impact on human health, society, and the economy. These include the following:Disease surveillance and laboratory capacity: Monitoring an infectious disease’s incidence and spread, as well as testing samples to confirm its presence is critical in the early detection of outbreaks.Strategic medical stockpiles: Stockpiling essential medical supplies, such as therapeutics, vaccines, and personal protective equipment, ensures that healthcare workers have the necessary resources to respond rapidly to an outbreak.Emergency preparedness planning: This involves developing plans to address disease outbreaks, including the activation of emergency response teams, isolation and quarantine measures, and communication strategies to the public.Infection control measures: These involve implementing measures to reduce the spread of infectious diseases. They include hand hygiene, the proper disposal of infectious waste, the proper use of personal protective equipment, social distancing, and the isolation/quarantine of infected individuals.Vaccination: Vaccination can play a key role in preventing and controlling the spread of infectious diseases. As seen with the COVID-19 pandemic, vaccines can mitigate a disease’s impacts and reduce pathogen transmission.Communication: Effective communication is critical in any pandemic response, as it helps to provide accurate information to the public and to prevent panic and the spread of misinformation.Research and development of new tools: Continuous support of research is needed for developing new countermeasures, such as vaccines and therapeutics, to prevent and control outbreaks of new and re-emerging pathogens.

While these strategies and tools are essential for global pandemic preparedness, there are limitations and gaps in the current system, such as the use of convalescent blood products [29]. Some of these challenges include the need for better surveillance and data analysis systems, more rapid and effective response mechanisms, and increased funding and resources to support preparedness and response activities.

PRT could be an infection control measure in the early stages of disease emergence, particularly when convalescent plasma is used as a therapy. Convalescent plasma as a passive immunotherapy has been used since the late 1800s to treat many infectious diseases [29]. It is also important to note, however, that transmissions with Zika, RRV, and WNV have occurred in the acute phase of illness even in the presence of low levels of neutralizing antibodies, suggesting that the effectiveness of these products may be somewhat limited [14]. During the COVID pandemic, by the spring of 2021, doctors in the United States had treated over 500,000 COVID-19 patients with convalescent plasma [30]. The use of these products before a complete understanding of viremia and the persistence of these agents in recovering patients raises concerns regarding the potential risk of disease transmission. These risks could be mitigated by the use of PRT. Implementing PRT early in disease outbreaks could minimize transmission risks associated with therapies such as convalescent plasma and potentially enhance blood transfusion safety.

## 3. Threats from Emerging Viruses

Emerging and re-emerging viruses have become a major concern for public health in more recent years. They can cause significant impacts on daily life, including social, economic, geopolitical, and health impacts [31]. The COVID-19 pandemic demonstrated how emerging threats can quickly disrupt the healthcare system [32], leading to widespread illness, death, and social and economic upheaval. In addition to SARS-CoV-2, mpox, Ebola virus, and hepatitis E virus have emerged in recent years and have posed significant threats to public health [33,34,35]. Of interest is the impact these viruses have on blood transfusion medicine. These pathogens are zoonotic and have the potential to cause severe illness and even death [36,37,38,39]. Therefore, understanding the transmission risk of emerging viral pathogens and implementing proactive mitigation strategies are critical to reducing the transmission risks of blood transfusions.

### 3.1. Mpox

Mpox, previously known as monkeypox, is a zoonotic viral pathogen that has recently re-emerged and quickly spread globally. As of April 2023, there have been over 87,000 reported cases [40]. The virus is a double-stranded DNA virus in the genus *Orthopoxvirus*. Its clinical symptoms may include rash, fever, chills, lymphadenopathy, headaches, and muscle aches [41]. The incubation period ranges from 5 to 21 days depending on the route of exposure [42]. The current mpox outbreak has been primary transmitted from human to human by sexual contact, fomite, and direct contact with infected lesions; respiratory modes have also been documented [43]. Rapid systemic spread and viremia in infected patients have been reported, raising concerns of potential blood-borne transmission, and many have questioned blood transfusion safety, though the primary route of transmission is skin-to-skin contact [44,45,46].

Numerous studies have reported the detection of mpox DNA in blood samples taken from infected patients using PCR [47,48,49,50,51,52,53,54,55,56,57,58,59]. Although the infectious dose required for blood-borne transmissions is unknown at this time, several case reports have raised the question of the hematogenous spread of the virus through accidental needle pricks with contaminated needles [60,61,62]. To determine the infectious potential of blood samples, researchers have correlated infectious viral titer with viral DNA levels in skin lesions and oropharyngeal swabs. Paran et al. determined that one hundred and seventy-two DNA copies correlate to one plaque forming unit (pfu) and suggested that samples with PCR values of less than four thousand three hundred copies/mL should be regarded as having no to minimal infectivity [63]. Given that the PCR values observed in blood samples from infected individuals exceed this amount, the likelihood of blood-borne transmission cannot be ruled out.

### 3.2. SARS-CoV-2

The emergence of a novel coronavirus, SARS-CoV-2, has led to a global spread of COVID-19 disease and continues to impact human health. SARS-CoV-2 is one of three major coronaviruses that have emerged in the last 20 years. The rapid emergence and spread of this new virus raised concerns about its potential impacts on blood transfusion medicine. Although its primary route of transmission is via respiratory spread, blood-borne transmission was theorized early in the pandemic [64]. The COVID-19 pandemic did demonstrate how the safety of blood transfusion products can be questioned in the face of a pandemic threat.

To date, there have been no known cases of transfusion transmission for the three recently emerged human coronaviruses, SARS-CoV, MERS-CoV, and SARS-CoV-2. Studies have been conducted to evaluate the potential risk of blood transmissions, and RNA has been detected in patients infected with SARS-CoV and MERS-CoV [65,66,67,68,69]. RNA was also detected in asymptomatic and symptomatic SARS-CoV-2 patients [70,71,72,73,74]. This includes the detection of RNA at the time of blood donation [75]. However, a case report from Asia describes an asymptomatic SARS-CoV-2-infected donor who did not transmit the virus to recipients of the donated units [76]. Furthermore, the infectious virus has not yet been isolated from RNA-positive blood in vitro [77,78,79,80,81] and in vivo models [82]. Experimental studies in a mouse model found that 10^4^–10^5^ plaque-forming units given intravenously are needed to develop an infection. However, such dose equivalents have not been detected in donated human blood samples, and the authors concluded that the risk of blood transmission is minimal [82]. Lastly, the prevalence of RNA in the blood from donors is low [83]. Therefore, the current risk level of SARS-CoV-2 transmission via blood transfusions is minimal.

Due to the uncertainty surrounding transmission risks early in the COVID-19 pandemic, blood donation centers implemented protocols to prevent infected people from donating and spreading the virus, both within donation centers via aerosols and through transfusion transmission. Unfortunately, this led to devasting impacts on the blood supply, causing a significant blood shortage. The World Health Organization estimated a 20–30% reduction in blood supply to all six of its regions. [84]. Additionally, the American Red Cross reported a 10% decline in the number of donations since the start of the pandemic [85]. In a recent meta-analysis, the decrease in blood donations during the COVID-19 pandemic was estimated at 38% on average, reaching 67% in some regions [86]. Even after lockdown restrictions had been lifted, blood services continued to observe lower whole blood donor availability [84,87]. Donation centers must now focus their efforts on regaining the willingness/intention to donate among individuals who deferred [88]. The COVID-19 pandemic highlights the short- and long-term impacts emerging viruses can have on blood transfusion medicine. PRT may be valuable in mitigating these impacts early in disease outbreaks, especially when the pathophysiology and transmission of the new pathogen are still unknown.

### 3.3. Ebola

Ebola viruses continue to re-emerge and cause significant outbreaks and epidemics in Africa. Belonging to the *Filoviridae* family, this highly infectious zoonotic virus can result in acute hemorrhagic fever and death. While a wide range of mammals and primates can act as hosts, fruit bats are thought to be the natural reservoirs [89]. The highest risk for human–human transmission is by direct contact with infected bodily fluids and tissues. However, the virus can also spread by fomites, droplets, and aerosols [90]. Its incubation period can be up to 21 days [91], which can make the containment of its spread more challenging during an outbreak. Mortality rates can range from 25 to 90% during outbreaks [92]. Due to the high mortality rates, as well as the various routes of transmission, the CDC lists ebolaviruses as a Category A bioterrorism agent and requires biosafety level 4 facilities to safely handle infected samples.

The virulence in human patients varies depending on the viral strain, with Ebola Zaire being the most fatal. Patients are considered infectious at the time of presenting clinical illness, which includes flu-like symptoms, rash, hemorrhaging, gastrointestinal signs, myalgia, and fatigue [93]. In severe cases, viremia can be very high in the blood [94,95]. Therefore, there is a risk of blood-borne transmission, although so far there have been no reported cases. There have been reported cases of bloodborne transmission, including direct contact with bodily fluids, needles [96], and shared needle injections [97]. Blood transfusions continue to be a valuable treatment for infected patients [98], and therefore, there is a high demand for transfusion products during outbreaks. It is unlikely that, during an outbreak, viremic patients infected with Ebola virus (EBOV) would be allowed to donate blood since viremia is associated with symptomatic disease. However, infectious viruses and RNA have been detected in various bodily fluids several months after the resolution of clinical disease [99]. Furthermore, the infectious dose is believed to be about 10 viral particles [100], which is very low. Therefore, precautions must be taken to avoid potential transfusion transmissions.

### 3.4. Hepatitis E

Hepatitis E virus (HEV) is a hepatotropic virus that causes acute hepatitis, which typically is self-limiting in most adults but can progress to chronic infection in immunocompromised individuals. This virus was first identified in 1983 during an outbreak of unexplained hepatitis in Soviet soldiers returning from Afghanistan [101]. HEV is classified into four genotypes: genotypes 1, 2, 3, and 4. Genotypes 1 and 2 are primarily found in developing countries in Africa and Asia and have been associated with waterborne epidemics. Genotypes 3 and 4 can infect both humans and pigs. Genotype 3 has a global distribution, while genotype 4 has been primarily detected in Asia [102].

The incubation period of HEV ranges from 2 to 8 weeks, and clinical symptoms include fever, nausea, abdominal pain, jaundice, and malaise [103]. HEV is typically spread by fecal–oral transmission in developing countries. However, other modes of transmission have been reported, such as vertical transmission [104,105,106,107,108], zoonosis from contact with infected pigs, consuming undercooked pig meat, or environmental contamination from pig slurry [109].

Of concern are the increasing reports of blood transfusion transmissions. Donors are typically asymptomatic at the time of donation and are not routinely screened for HEV infection. However, there have been reports of HEV infections in transfused patients, leading to investigations that identified transfused patients positive for antibodies or RNA from HEV infections [110,111,112]. Endemic areas such as India are at an increased risk of transfusion transmissions [113], whereas an increase in HEV cases in the general population in Europe has been reported by the ECDC. HEV infection transmissions have also been reported in Japan [114], China [115], Germany [116], England [13], the Netherlands [117], Scotland, and Austria [118]. Transfusion-transmitted HEV infections range in their clinical outcomes from short viremias to chronic infections, as seen in cases from England [13]. The infectious dose necessary for transfusion transmission is unknown, but it is believed to be a low dose and is at the limit of detection by PCR [119]. As a non-enveloped virus, HEV poses extra challenges to demonstrate complete inactivation. A report from Hauser et al. described two cases of HEV transmission by blood products treated by the psoralen + UVA-based PRT method [120]. HEV demonstrates the need for effective strategies to mitigate blood transfusion transmissions globally for emerging viruses.

These emerging viral pathogens underscore the potential impact on blood transfusion medicine. This in turn has significant short-term and long-term consequences on blood safety and life-saving treatments. It also emphasizes the importance of continuing to develop effective strategies to maintain the safety and quality of blood transfusions to prevent the spread of emerging infectious diseases.

## 4. Riboflavin + UV Light Pathogen Reduction Technology

The effectiveness of the Mirasol PRT System against a broad range of pathogens has been previously described for multiple blood product types [121]. Designed to be a pathogen-agnostic system, the technology has a demonstrated ability to reduce the infectious pathogen load of viruses, bacteria, and parasites in plasma, platelets, and whole blood products, including against emerging diseases for which the development of diagnostic tests may lag behind the emergence of the pathogen. Of importance, the Mirasol PRT System has been demonstrated to be effective against mpox, SARS-CoV-2, Ebola, and HEV (Table 1).

### 4.1. Mpox

To evaluate the effectiveness of the Mirasol PRT System against the mpox virus that emerged in 2022, plasma and whole blood products (n = three of each) were inoculated with the mpox virus (USA_2003), with pre-treatment titers of 3.50 and 3.08 log10 pfu/mL, respectively [127]. These pre-treatment viral titers were clinically relevant based on the amount of the virus detected in the patients infected with the mpox virus. All products were treated with the Mirasol System as per the manufacturer’s instructions for each product type, and for all products, the post-treatment titer was below the limit of detection.

### 4.2. SARS-CoV-2

Before it was known whether SARS-CoV-2 could be transmitted via blood, several studies were performed to assess the ability of the Mirasol PRT System to reduce the viral load in blood products. In one study, plasma and whole blood products were inoculated with 3–4 log10 pfu/mL of SARS-CoV-2 virus (USA-WA1/2020) [124]. The rate of viral inactivation was evaluated in the plasma products with energy doses of 30, 60, and 100% of the target dose delivered, and the post-treatment titers were measured. The whole blood products were treated as per the manufacturer’s instructions with 100% of the target UV dose. The viral titers reached the limit of detection at 60% of the target energy dose in the plasma products, while the treatment of the whole blood products yielded an average viral reduction of 3.30 log10, demonstrating that Mirasol is efficacious in inactivating SARS-CoV-2 in plasma and whole blood products.

These data were further bolstered in a second study that reported the results from a Mirasol treatment of both plasma and platelet products in plasma [125]. The pre-treatment titers for all the products were greater than 4.3 log10 pfu/mL, and, as in the first study, the treatment resulted in no detectable virus plaques, meaning that the remaining infectious titers were below the limit of detection of the standard plaque assay.

Because convalescent plasma collected from patients who had recovered from SARS-CoV-2 infection was utilized in the early days of the pandemic to treat patients with active disease, a third study was performed to evaluate whether the neutralizing antibodies present in SARS-CoV-2 convalescent plasma products would be preserved following Mirasol PRT treatment [128]. Plasma products collected from known SARS-CoV-2 convalescent donors were collected, and pre-treatment neutralizing antibody titers were determined by both a plaque reduction neutralization test against the live SARS-CoV-2 virus, as well as a pseudovirus reporter viral particle neutralization (RVPN) assay. Spike protein receptor-binding domain (RBD) and subunits S1 and S2 were also evaluated using an enzyme-linked immunosorbent assay (ELISA). Minimal effects to the measured antibodies were demonstrated in all the assays, suggesting that the Mirasol System is effective in reducing the viral burden in blood products while simultaneously conserving the therapeutic benefits of convalescent plasma components.

### 4.3. Ebola

The Ebola epidemic that originated in Guinea, West Africa in 2013 lasted for over two years and claimed over 10,000 lives, spreading not only within Africa but to other continents as well, as health care workers working on the front lines traveled back to their home countries and then tested positive for infection. At the time, there was no approved vaccine for EBOV, and the use of convalescent plasma was implemented to combat the disease.

Cap et al. (2016) evaluated the effectiveness of a Mirasol PRT treatment of plasma and whole blood for the inactivation of Ebola virus. The investigators reported that the UV+ riboflavin treatment reduced the EBOV titers to non-detectable levels in both nonhuman primate serum (≥2.8- to ≥3.2-log reduction) and human whole blood (≥3.0-log reduction) without decreasing the protective antibody titers in human plasma [123].

### 4.4. Hepatitis E

Initially believed to only be transmitted orally [129,130], HEV is now understood to be also transmitted by blood transfusion [131] and can cause severe hepatitis. HEV is classified into four genotypes (G1–G4), with G3 being the most widely distributed globally. In order to evaluate the ability of the Mirasol System to be used as a tool to prevent transfusion transmissions of HEV, Owada et al. (2014) reported that they used plasma and serum specimens collected and cultured from HEV-RNA-positive patients to produce the JRC-HE3 strain for G3 and a UA1 strain for G4 [122]. These authors further observed that the Mirasol PRT system achieved a >3 log inactivation for the JRC-HE3 strain and a >2 log inactivation for the UA1 strain of HEV. They concluded that the Mirasol PRT system “inactivated greater than 2 to 3 logs of live HEV in PLTs and can potentially be used to lower the possibility of bloodborne HEV transmission”.

## 5. Discussion

Over the last two decades, the world has experienced how changes in the environment, society, and human behavior have resulted in the emergence or re-emergence of infectious agents. Some of these pathogens were able to be transmitted by blood transfusions and represented an unknown risk at the time of their appearance [18,132,133]. The standard safety measures in place, such as donor selection and donation screening, were not immediately effective in reducing the risk of transfusion. Moreover, a continued strategy of additional screening and donor deferral for microbial safety is unlikely to be sustainable in the future, and in some cases, this approach has proven to be extremely costly, due to very low positive detection rates of regional infections [16,17]. Introducing new tests at a time when transfusion transmission is uncertain and unknown also has the potential to introduce costs into already financially strained services that later prove to be unwarranted and unnecessary.

Pathogen reduction technologies for labile blood components, especially for platelet concentrates, have been shown to decrease the risk of bacterial transmission in countries where these technologies have been nationally applied [24,134]. A milestone trial that investigated PRT’s capability to decrease malaria transmission by blood in a malaria-endemic country, showed a significant reduction (87%) in transmission events in the patient group that received the riboflavin + UV light-treated whole blood [25].

Pathogen reduction technologies for labile components are in their first-generation release and cannot completely eliminate infectious risks [120], as seen by some transmission cases despite PI/PRT treatment [120,135]. Pathogen reduction performance varies due to the physical/structural/genetic composition of the pathogen. Some technologies are unable to inactivate non-enveloped viruses while some are more effective against these agents [120,136,137]. Nevertheless, the treatment adds a new layer of safety that might contribute to closing the window period of detection for tested viruses such as HIV, HBV, and HCV and reduce the transmission risk as compared to that of untreated components [138,139]. Cost-effectiveness investigations of these technologies have shown different outcomes depending on the epidemiological characteristics of the population aimed to be served by the technology [140,141,142]. The use of PRT methods also has been demonstrated to require increased numbers and frequency of transfusions in order to compensate for reduced levels of therapeutic efficacy resulting from damage to components during treatment [143,144]. Clearly, the decisions to implement or not implement PRT methods must depend on the specifics of the disease’s outbreak, transfusion risk endemicity, cost considerations, and clinical considerations that are not uniform geographically or temporally.

Unquestionably, the success of this blood transfusion intervention will depend on the ability of manufacturers to develop easy-to-use technologies to treat whole blood for componentization without the use of toxic compounds that may pose risks to individuals and the environment. The proof of this concept has been delivered in a pilot study in Russia, where pediatric patients have been transfused with red cell concentrates produced by the fractionation of riboflavin + UV light-treated whole blood, showing that the performance of these components remained clinically acceptable [27]. Moreover, two patients in this group also received plasma and platelet concentrates treated with the same PR technology, showing the positive therapeutic effect of these interventions after each transfusion [145]. This was the first documented case of universal component pathogen reduction being employed in a clinical setting.

As changes in climate, transportation, and social dynamics continue to drive the emergence of new diseases in human populations, innovation and technology must advance our ability to respond to and address infectious disease threats. The interdependence of blood safety and availability with general human population health is based on the fact that transfusion products are derived from human donors. Their susceptibility to emerging disease threats which may alter the integrity of these products and pose a risk for transfusion recipients will continue to be a factor driving blood availability for patients who require transfusion support. New methods designed to reduce or eliminate infectivity in blood products through proactive pathogen reduction methods have been developed and are being implemented globally for blood components, including plasma, platelets, red cells, and whole blood. The ability of these processes to address emerging disease threats has been tested in real time as these agents are emerging and even before the full threat of transfusion transmission is known. Given the likelihood that diseases will continue to emerge, investments in the exploration of new ways to implement these methods in logistically practical and cost-effective ways for all components of blood seem both prudent and warranted.

## Figures and Tables

**Table 1 pathogens-12-00911-t001:** Blood product types used to evaluate PRT inactivation of viruses of interest.

Viral Pathogen	Platelets	Plasma	Whole Blood	Reference
Hepatitis E	✓	✓	Not tested	[122]
Ebola	Not tested	✓	✓	[123]
SARS-CoV-2	✓	✓	✓	[124,125,126]
Mpox	Not tested	✓	✓	[127]

✓ indicates blood product type has been tested.

## Data Availability

All data utilized in this review were obtained from publicly available records and published data which are included in the reference section of the manuscript with citations.

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
