# Peer review of "Emerging Pathogen Threats in Transfusion Medicine: Improving Safety and Confidence with Pathogen Reduction Technologies"

_pathogens, 2023, doi:10.3390/pathogens12070911_

Round 1

Reviewer 1 Report

This is a well written article that predominately focusses on the experimental results of pathogen reduction for selected infectious agents, the majority of which have not been shown to be transmitted by transfusion and in particular one that is extremely unlikely to be.  It does not critically appraise pathogen reduction’s role in emerging threats and there are few limitations or a rounded argument including risk-based decision making including cost-effectiveness, although it is mentioned in passing. My opinion is it may be suitable for publication with a major revision that presents a more rounded argument including the limitations and cost effectiveness of PR.

Title; it states the review is on the role of pathogen reduction. However, this review does not demonstrate the role of pathogen reduction as there is no discussion on a complete risk assessment but as per the abstract is a review of real time evaluation for emerging infectious diseases using pathogen reduction focussing on effectiveness.

Abstract

Page 1 line 14-15 suggest scientific wording change

Page 1 line 17-18 vector for spreading disease sounds very emotive, suggest wording change.

Page 1 line 21-22 it states pathogen reduction can significantly reduce the likelihood of transfusion transmission for emerging agents as they emerge. That is only in the event that they are transfusion-transmitted, and many are not or are a negligible  risk with existing precautions.

Introduction

Page 2; Line 52-54. The statement reads ‘Along with the classical blood-borne pathogens, HIV, HBV, HCV, arbovirus transmission 52 through blood transfusion has been increasingly reported in the past 20 years [12,13].’

The references are suitable for the BBV and hepatitis E and no not reference arboviruses at all so not appropriate.

The next statement about WNV and Dengue reads quite weakly. They are more than suspected, there is strong evidence that they are transmitted by transmission.

Page 3 line59-61 states that tests rely on NAT but that is not true of syphilis which is tested via serological screening. For syphilis ceasing whole blood transfusion and direct vein to vein ie modern blood processing techniques have played a role

Page 3 64-64 ‘highlights the inadequacy of reactive prevention strategies to quickly address emergent needs in a timely way”.  Fortunately there have not been any examples of repeats of the HIV and HCV epidemics and testing development has come a long way. This does not acknowledge that.

Page 2 line 77 states the processed red cell concentrate has proven to be safe and effective using reference 21 but the actual reference states in the conclusion ‘It is a promising method to increase safety without affecting clinical efficacy and a RCT including more patients should follow this pilot study to confirm its results. Therefore, the wording should be revised.

Page 2 line 86-89 it states the article will review strategies to respond to infectious threats with a special focus on transfusion medicine plus the next sentence. This article does not do this in any great depth and the focus is on PR. Therefore, this should be reworded. From moreover, it is an accurate description of what the article tries to achieve.

The section pandemic preparedness for infectious diseases is about a comprehensive pandemic response and does not go into specifics for blood safety threats.

Section 3.1 Mpox

It should be called mpox and previously known as monkeypox or monkeypox virus.

Page 4 lines 153-154. I believe this is a misinterpretation of the needlestick data. Mpox is transmitted through skin-to-skin contact. The needlestick data demonstrates that using a contaminated object, in some cases deroofing mpox lesions, from an infected persons skin and piecing the skin of the health care worker with the mpox contaminated object demonstrates transmission ie skin transmission from a contaminated sharp object. This does not demonstrate hematogenous spread. Clearly it does not rule it out though.

3.2 SARs CoV-2

Line 167-168. Suggest reword. We now know implies there was some question about how SARS-CoV-2 was spread. It was known very early on that this was via the respiratory route. There was a question about the minor potential of transfusion-transmission but even very early on it was acknowledged that whilst it could not be ruled out it was a negligible risk as other similar infections have not been shown to be transfusion-transmitted.

Lines 184-188. This states due to the uncertainty of the transmission risk that blood donation centres implemented protocols to prevent infected people from donation. This does not specific what the authors are referring to either respiratory transmission in centre or transfusion transmission or both. In reality it was both but because the article is about transfusion-transmission it implies the later. The reference 70 references a US article that implies in the US it was because blood drives were cancelled because businesses and schools etc were closed. This is a very important distinction. If the risk of transmission is overwhelmingly via the respiratory route and even if it is rarely transfusion-transmitted then pathogen reduction is not going to solve the issue  in a pandemic  and the bigger concern will always  be respiratory transmission in blood donation centres.

Ebola

With Ebola, if it was a significant risk similar to SARS-CoV-2 there should be  concern about person to person spread during blood donation collection. Pathogen reduction as a solution does not address this.

HEV

Line 237-238. Many countries have introduced HEV screening. For some that have not this is because it has demonstrated to be not warranted based on risk. That statement therefore needs changing. Of concern are the increasing reports of blood transfusion transmission.

Section 4 Riboflavin.

Line 263-264 It refers to Table 1 and states it has been demonstrated to be effective against… What does effective mean?  100% success or a reduction. Reference 107 for HEV demonstrates a log reduction 2-3 logs and the last paragraph concludes further studies are needed under clinical conditions. The linking to Table 1 and the word effective may imply the ticks are that it has been proven effective when are they actually representing that PR has been trialled for these components?

It is notable that only 1 of the 4 agents discussed has been demonstrated to be transfusion-transmitted but the so what factor has not ben discussed and the fact that close contact or respiratory transmission in centre would be a greater risk if infectious people were coming to donate.

Section 4.4 HIV line 317 it states HEV was initially believed to be transmitted orally- this is a true statement and the overwhelming majority of transmissions. Do you mean was not known to be transfusion-transmitted?

Discussion

First paragraph lines 328-338. This  is very vague and not refenced. What pathogens are the authors referring to?

If testing is very costly that is because formal risk analysis including CEA have not been implemented before introduction and the same can be said in many cases for PR.

very well written a couple of typos and hepatitis E should be lower case.

Author Response

Reviewer 1:

This is a well written article that predominately focusses on the experimental results of pathogen reduction for selected infectious agents, the majority of which have not been shown to be transmitted by transfusion and in particular one that is extremely unlikely to be.  It does not critically appraise pathogen reduction’s role in emerging threats and there are few limitations or a rounded argument including risk-based decision making including cost-effectiveness, although it is mentioned in passing. My opinion is it may be suitable for publication with a major revision that presents a more rounded argument including the limitations and cost effectiveness of PR. 

Title; it states the review is on the role of pathogen reduction. However, this review does not demonstrate the role of pathogen reduction as there is no discussion on a complete risk assessment but as per the abstract is a review of real time evaluation for emerging infectious diseases using pathogen reduction focussing on effectiveness.

Response: The authors changed the title to: “Emerging Pathogen Threats in Transfusion Medicine: improving safety and confidence with Pathogen Reduction Technologies”

Abstract

Page 1 line 14-15 suggest scientific wording change

Response: Changes have been made, as suggested by the reviewer

Page 1 line 17-18 vector for spreading disease sounds very emotive, suggest wording change.

Response: Changes have been made, as suggested by the reviewer.

Page 1 line 21-22 it states pathogen reduction can significantly reduce the likelihood of transfusion transmission for emerging agents as they emerge. That is only in the event that they are transfusion-transmitted, and many are not or are a negligible risk with existing precautions.

Response: That is correct but this fact is not known at the moment the emergent pathogen is identified and already at that time, a protective measure is in place.

Introduction

Page 2; Line 52-54. The statement reads ‘Along with the classical blood-borne pathogens, HIV, HBV, HCV, arbovirus transmission 52 through blood transfusion has been increasingly reported in the past 20 years [12,13].’

The references are suitable for the BBV and hepatitis E and no not reference arboviruses at all so not appropriate.

The next statement about WNV and Dengue reads quite weakly. They are more than suspected, there is strong evidence that they are transmitted by transmission.

Response: Corrections and changes have been made in accordance to the reviewers' suggestions.

Page 3 line59-61 states that tests rely on NAT but that is not true of syphilis which is tested via serological screening. For syphilis ceasing whole blood transfusion and direct vein to vein ie modern blood processing techniques have played a role

Response: Changes have been made to the text to address the reviewer’s comments.

Page 3 64-64 ‘highlights the inadequacy of reactive prevention strategies to quickly address emergent needs in a timely way”.  Fortunately there have not been any examples of repeats of the HIV and HCV epidemics and testing development has come a long way. This does not acknowledge that. 

Response: Changes have been made to the text to give credit to the responsive measures that enabled increase in blood safety in the past decades. A new reference has been added to the text.

Page 2 line 77 states the processed red cell concentrate has proven to be safe and effective using reference 21 but the actual reference states in the conclusion ‘It is a promising method to increase safety without affecting clinical efficacy and a RCT including more patients should follow this pilot study to confirm its results. Therefore, the wording should be revised.

Response: Wording has been revised accordingly.

Page 2 line 86-89 it states the article will review strategies to respond to infectious threats with a special focus on transfusion medicine plus the next sentence. This article does not do this in any great depth and the focus is on PR. Therefore, this should be reworded. From moreover, it is an accurate description of what the article tries to achieve.

Response: The paragraph has been changed to better clarify the objective of this manuscript.

The section pandemic preparedness for infectious diseases is about a comprehensive pandemic response and does not go into specifics for blood safety threats.

Response: The section on pandemic preparedness was expanded to discuss the role of PRT as a tool in disease control during early stages of disease spread. We also added a brief discussion of PRT and blood safety threats when convalescent plasma is used as a treatment in the face of disease outbreaks.

Section 3.1 Mpox

It should be called mpox and previously known as monkeypox or monkeypox virus.

Response: Line 170 has been updated, as suggested by the reviewer.

Page 4 lines 153-154. I believe this is a misinterpretation of the needlestick data. Mpox is transmitted through skin-to-skin contact. The needlestick data demonstrates that using a contaminated object, in some cases deroofing mpox lesions, from an infected persons skin and piecing the skin of the health care worker with the mpox contaminated object demonstrates transmission ie skin transmission from a contaminated sharp object. This does not demonstrate hematogenous spread. Clearly it does not rule it out though.

Response: Thank you for identifying the confusion. The reviewer is correct that there was potential skin transmission from a fomite (contaminated sharp object). However, as the reviewer pointed out, hematogenous spread cannot be ruled out.  We have updated the sentence to clarify the potential route of transmission. ”Although the infectious dose required for blood-borne transmission is unknown at this time, several case reports have raised the question of hematogenous spread of the virus through accidental needle sticks with contaminated needles.”

3.2 SARs CoV-2

Line 167-168. Suggest reword. We now know implies there was some question about how SARS-CoV-2 was spread. It was known very early on that this was via the respiratory route. There was a question about the minor potential of transfusion-transmission but even very early on it was acknowledged that whilst it could not be ruled out it was a negligible risk as other similar infections have not been shown to be transfusion-transmitted. 

Response: The sentence was reworded to emphasize airborne transmission as the primary route. The sentence was changed to “Although the primary route of transmission is via respiratory spread, blood-borne transmission was theorized early in the pandemic”.

Lines 184-188. This states due to the uncertainty of the transmission risk that blood donation centres implemented protocols to prevent infected people from donation. This does not specific what the authors are referring to either respiratory transmission in centre or transfusion transmission or both. In reality it was both but because the article is about transfusion-transmission it implies the later. The reference 70 references a US article that implies in the US it was because blood drives were cancelled because businesses and schools etc were closed. This is a very important distinction. If the risk of transmission is overwhelmingly via the respiratory route and even if it is rarely transfusion-transmitted then pathogen reduction is not going to solve the issue  in a pandemic  and the bigger concern will always  be respiratory transmission in blood donation centres.

Response: Thank you for the comment. We did intend to discuss both respiratory transmission in centers and transfusion transmission. We have changed the wording in the first sentence to clarify that. We also clarified that PRT would be best used early in a disease outbreak when transmission of the pathogen is unknown.

Ebola

With Ebola, if it was a significant risk similar to SARS-CoV-2 there should be  concern about person to person spread during blood donation collection. Pathogen reduction as a solution does not address this.

Response: Thank you for the comment. Yes, there is concern of person-to-person transmission of Ebola during blood donation collection. And the reviewer is correct that PRT would not address that. However, we discuss the concern that Ebola is transmitted by bodily fluids and that the infectious dose is low. Therefore, precautions should be taken with transfused blood products. PRT can be valuable in improving blood transfusion safety in this setting.

HEV

Line 237-238. Many countries have introduced HEV screening. For some that have not this is because it has demonstrated to be not warranted based on risk. That statement therefore needs changing. Of concern are the increasing reports of blood transfusion transmission. 

Response: Thank you for this comment. We are clearly referring to the increase of HEV reported cases to the ECDC.

Section 4 Riboflavin.

Line 263-264 It refers to Table 1 and states it has been demonstrated to be effective against… What does effective mean?  100% success or a reduction. Reference 107 for HEV demonstrates a log reduction 2-3 logs and the last paragraph concludes further studies are needed under clinical conditions. The linking to Table 1 and the word effective may imply the ticks are that it has been proven effective when are they actually representing that PR has been trialed for these components?

Response: In each case, the levels of inactivation for the agents tested has been at the limit of detection of the assays used.  These assays measure infectivity, not copy number.  In general, more copies are present per infectious agent, suggesting a margin that is difficult to define because the infectious dose per copy number for each agent is not known.

It is notable that only 1 of the 4 agents discussed has been demonstrated to be transfusion-transmitted but the so what factor has not been discussed and the fact that close contact or respiratory transmission in centre would be a greater risk if infectious people were coming to donate. 

Response: These four agents represent recent emerging pathogens with known (HEV and Ebola), suspected (MPox) and ultimately no (SARS-2) blood transmission route.  In every case PRT was able to demonstrate effectiveness in reducing viral titers in blood, thus correspondingly reducing real or perceived risks to blood product integrity.

Concerning risks of transmission through aerosol and blood transmission has been clearly stated in section 3.2

Section 4.4 HIV line 317 it states HEV was initially believed to be transmitted orally- this is a true statement and the overwhelming majority of transmissions. Do you mean was not known to be transfusion-transmitted?

Response: We added the following: “Initially believed to only be transmitted orally, HEV is now understood to be transmitted by blood transfusion and can cause severe hepatitis disease.”

Discussion

First paragraph lines 328-338. This  is very vague and not refenced. What pathogens are the authors referring to?

Response: The authors are referring firsthand on the WNV epidemics in the USA.

If testing is very costly that is because formal risk analysis including CEA have not been implemented before introduction and the same can be said in many cases for PR.Response:

Response: That is correct. The authors refer to the three CEA and CUA done for PRT based on Riboflavin + UV light treatment. CEA will be strongly related to countries infectious diseases epidemiology.

Reviewer 2 Report

This review describes the most recent outbreaks of viral pathogens with potential relevance for blood transfusion safety. In addition, it highlights the role of pathogen reduction as preventive measures against the threats from emerging viruses and shortly addresses important aspects relevant for implementation of this technology.

There is no doubt that implementation of the pathogen reduction technology would provide an additional safety layer for transfusion medicine. However, the fact that the available technologies differ in their applicability and inactivation capacity requires a more differentiated discussion. Although this review focuses on the riboflavin plus UV light pathogen reduction technology, this method should be put into the context of other existing methods (UVA plus amotosalen, UVC). A short summary of the different mechanisms of action, the blood product types that can be treated, and the efficacies for the different emerging pathogens should be provided.

Author Response

Reviewer 2:

This review describes the most recent outbreaks of viral pathogens with potential relevance for blood transfusion safety. In addition, it highlights the role of pathogen reduction as preventive measures against the threats from emerging viruses and shortly addresses important aspects relevant for implementation of this technology. 

There is no doubt that implementation of the pathogen reduction technology would provide an additional safety layer for transfusion medicine. However, the fact that the available technologies differ in their applicability and inactivation capacity requires a more differentiated discussion. Although this review focuses on the riboflavin plus UV light pathogen reduction technology, this method should be put into the context of other existing methods (UVA plus amotosalen, UVC). A short summary of the different mechanisms of action, the blood product types that can be treated, and the efficacies for the different emerging pathogens should be provided.

Response: Thank you for this comment. The authors included more specific information about the range of PRT methods available to date.

Reviewer 3 Report

In the introduction I miss a good explanation/summary about the different PRT available/developed. The readers perhaps don't understand or are not experts on the subject, so it would be advisable to introduce all the possible technologies and products inactivated, referred to review papers.

At the end of the introduction authors state that they are going to review strategies to... withnspecial focus in transfusion medicine. That's not true as lon as it is focused in PRT and only in riboflavin, so thhis sentence should be stated in another way.

Page 4, section 3.2

line 187 reference 70 is not for global supply is for Africa

line 190 it says 76% is a mistake (67%)

4.1 section

It not clear if it is something described in a previous paper (no reference) or if they are explaining something new. If the latter is true it should be explained with more detail, describing log reduction, etc. is

4,2 section

The same as section 4.1 

also they should describe what a high level efficacy is, for example

"While there are no guidelines defining the needed inactivation efficacy of PRT, LRFs of 4.0 log are generally considered the minimum requirement for viruses and parasites based on regulatory standards per the Committee for Human Medicinal Products."

Nahler G. Committee for Proprietary Medicinal Products (CPMP). In: Dictionary of pharmaceutical medicine. Vienna: Springer; 2009.

"However, requirements for labile blood components may differ, and, ultimately, the demonstrated LRF attained by PRT will be relevant only to define the extent to which other procedures (tests and deferrals) will need to be used in tandem with PRT."

Goodrich RP, Custer B, Keil S, et al. Defining "adequate" pathogen reduction performance for transfused blood components. Transfusion 2010;50:1827-37.

Lanteri MC, Santa-Maria F, Laughhunn A, Girard YA, Picard-Maureau M, Payrat JM, Irsch J, Stassinopoulos A, Bringmann P. Inactivation of a broad spectrum of viruses and parasites by photochemical treatment of plasma and platelets using amotosalen and ultraviolet A light. Transfusion. 2020 Jun;60(6):1319-1331. doi: 10.1111/trf.15807. Epub 2020 Apr 24. PMID: 32333396; PMCID: PMC7317863.

Through all the paper I miss some references in some places:

Introduction:

Page 2, line 69. Authors use reference 16, a reference focused on Mirasol use when they are talking about all systems, use a review on PRT.

Line 75, authors are talking about Europe and North America and the reference is about Europe

Page 3 section 3 (threats...)  line 135 after public health a reference is missing

Page 3 section 3 line 136 after transf medicine a reference is missing

line 137 after even death a reference is missing

PAge number 6 sectio 4

In the first paragraph I miss references after ...product types; .... after emergence oh the pathogen

PAge 8 

second paragraph lines 347-354 I think that there are better review references

Author Response

Reviewer 3:

In the introduction I miss a good explanation/summary about the different PRT available/developed. The readers perhaps don't understand or are not experts on the subject, so it would be advisable to introduce all the possible technologies and products inactivated, referred to review papers.

Response: Indeed; we have now included a short description of the alternative PR technologies.

At the end of the introduction authors state that they are going to review strategies to... withnspecial focus in transfusion medicine. That's not true as lon as it is focused in PRT and only in riboflavin, so thhis sentence should be stated in another way.

Response: Thank you for the comment. The authors redrafted this paragraph to better describe the purpose of this manuscript.

Page 4, section 3.2

line 187 reference 70 is not for global supply is for Africa

Response:  Reference 70 states “WHO estimated that the COVID-19 pandemic caused 20% to 30% reduction of blood supply in all its six regions…” To clarify, we changed the statement to say “ The World Health Organization estimated a 20%–30% reduction in blood supply to all six of its regions.

line 190 it says 76% is a mistake (67%)

Response: Thank you for catching this error. We have corrected the value from 76% to 67% in the manuscript.

4.1 section

It not clear if it is something described in a previous paper (no reference) or if they are explaining something new. If the latter is true it should be explained with more detail, describing log reduction, etc. is

Response: We have added references.

4,2 section

The same as section 4.1 

Response: We have added references.

also they should describe what a high level efficacy is, for example:

"While there are no guidelines defining the needed inactivation efficacy of PRT, LRFs of ≥4.0 log are generally considered the minimum requirement for viruses and parasites based on regulatory standards per the Committee for Human Medicinal Products."

Nahler G. Committee for Proprietary Medicinal Products (CPMP). In: Dictionary of pharmaceutical medicine. Vienna: Springer; 2009.

"However, requirements for labile blood components may differ, and, ultimately, the demonstrated LRF attained by PRT will be relevant only to define the extent to which other procedures (tests and deferrals) will need to be used in tandem with PRT."

Goodrich RP, Custer B, Keil S, et al. Defining "adequate" pathogen reduction performance for transfused blood components. Transfusion 2010;50:1827-37.

Lanteri MC, Santa-Maria F, Laughhunn A, Girard YA, Picard-Maureau M, Payrat JM, Irsch J,

Stassinopoulos A, Bringmann P. Inactivation of a broad spectrum of viruses and parasites by photochemical treatment of plasma and platelets using amotosalen and ultraviolet A light. Transfusion. 2020 Jun;60(6):1319-1331. doi: 10.1111/trf.15807. Epub 2020 Apr 24. PMID: 32333396; PMCID: PMC7317863.

Response: We have updated the statement to say that Mirasol is efficacious against SARS-CoV-2.

Through all the paper I miss some references in some places:

Introduction:

Page 2, line 69. Authors use reference 16, a reference focused on Mirasol use when they are talking about all systems, use a review on PRT.

Response: Thank you for your comment. The authors added two new references that discuss all different PR technologies and their specific features.

Line 75, authors are talking about Europe and North America and the reference is about Europe

Response. Authors added a reference about the PRT experience in the USA.

Page 3 section 3 (threats...)  line 135 after public health a reference is missing.

Response: We have added references.

Page 3 section 3 line 136 after transf medicine a reference is missing.

Response: The authors selected these four agents to represent recent emerging pathogens with known (HEV and Ebola), suspected (MPox) and ultimately no (SARS-2) blood transmission route.  In every case PRT was able to demonstrate effectiveness in reducing viral titers in blood, thus correspondingly reducing real or perceived risks to blood product integrity.

line 137 after even death a reference is missing

Response: We have added references.

PAge number 6 sectio 4

In the first paragraph I miss references after ...product types; .... after emergence oh the pathogen

PAge 8 

Response: We have added references.

second paragraph lines 347-354 I think that there are better review references

Response: The authors added a reference from the USA showing no TTI with PRT-treated PLTs

Round 2

Reviewer 1 Report

Thank you for your revisions. Whilst the manuscript is improved I think it needs further work.

I should have been more specific about the abstract.

I would suggest changing the words ‘increasingly problematic’ to something like a ‘a potential risk’

Instead of ‘to the spreading of diseases’ I would reword to ‘infectious disease transmission’.

Page 3 and convalescent plasma;  I would not say convalescent plasma is often used as a therapeutic. It has been in Ebola and for COVID and the evidence for efficacy. those is not yet there. I would rephrase to ‘has been used as a potential therapeutic until…”

Page 4 line 162. It states PRT is a valuable infection control measure in the early stages of disease emergency particularly when convalescent plasma is used as a therapy.

I  disagree- it has the potential to be a valuable infection control measure if the agent is transfusion-transmitted in the convalescent phase of the illness but that has not demonstrated to be the case for acute illnesses. Even when infectious diseases are transfusion-transmitted it is generally in the acute phase before neutralising antibodies are produced.

Therefore it should be changed to  PRT could be and also in line 169 it states can be when it is could be if the agent is subsequently confirmed to be transfusion transmission. I would also change it to potentially enhance blood safety at a minimum.

However, to be a balanced article the points above should be highlighted- transmissions with Zika, RRV, WNV have all occurred in the acute phase of the illness and not when someone has recovered even if low level viraemia is present there are neutralising antibodies. Therefore, I think the focus on convalescent plasma is misplaced. I would be more concerned about acute asymptomatic infections than a donor confirmed to be convalescent. I understand that during a pandemic as your new title suggests PR provides confidence when there is uncertainty. However, I do not agree that should be the sole reason to use it.

Page 4, Line 178 hepatitis E should be lower case

3.1 Title Mpox or Monkeypox virus

Agree with the change but to be balanced some note of skin to skin transmission should be raised.

4.3 Ebola it states for EBOV convalescent plasma was one of the only effective treatments to combat the disease but that is not correct from my understanding. There are no randomised clinical trials confirming convalescent plasma was an effective treatment for Ebola.

4.4 Hepatitis E-line 371, you do not need the word disease on the end of hepatitis.

Comment on 1 in 4 demonstrated to be TT (HEV) and the response is that Ebola is blood transmission risk and mpox is suspected. Whether something can be transmitted via body fluids is completely different to whether something can be transmitted via a blood transfusion from an asymptomatic person with modern blood storage techniques and when you are presenting an article on transfusion-transmission that distinction is very important. Transfusion transmission of Ebola has not been reported and that is a fact.

Page 9 line 402 it states PR are ‘currently the best strategy to pre-empt the next lethal blood transmitted epidemic. There is no evidence that this is true so this statement should be removed.

I strongly suggest that the limitations are clearly spelt out in the conclusion or one of the concluding paragraphs, otherwise I am still reading this as science plus a sell for the positive impacts of PR when this is a scientific journal. There is decrease in efficacy of products, introduction in many countries may not be cost justified if risk-based decision principles are adhered to. First world countries that have implemented PR have generally done so without considering cost and those that have considered it have generally not implemented it. Canada is the exception but based this on a probability of a HIV like infection occurring again and not current cost-effectiveness.

This is well written.

Author Response

Dear Reviewer

The authors want to thank you for the great review of the manuscript helping the authors to improve it considerably.

The answers to your comments have been followed by changes in the text of the manuscript in all cases (see document attached). The authors also reserved their right to respond to  one of reviewer's comments with an argumentation paragraph aiming for a constructive discussion. This paragraph is only addressed to the reviewer and not aimed to be included in the manuscript.

We hope the reviewer appreciates this dialogue.

Cordially

Marcia Cardoso, Izabela Ragan, Lindsay Hartson, Raymond P Goodrich

Round 3

Reviewer 1 Report

Thank you for your responses. My concerns have been addressed.  

There are a  couple of minor issues that can be addressed at the editing stage. 

For mpox there is still some cross over in referring to mpox or monkeypox in the text and for 4.1 My comment was intended as either refer to mpox or monkeypox virus not both.

For the new sentance on page 4 line 158 it should be clear that the effectiveness for CP products specifically is likely to be limited. As I don't think it is clear.

well written

Author Response

For mpox there is still some cross over in referring to mpox or monkeypox in the text and for 4.1 My comment was intended as either refer to mpox or monkeypox virus not both.

Response: The authors used in all citations the actual mpx abbreviation. The only time the word monkexpox is now used is in line 190 in the semi sentence: “Mpox, previously known as monkeypox,

For the new sentance on page 4 line 158 it should be clear that the effectiveness for CP products specifically is likely to be limited. As I don't think it is clear.

Response: The authors added a semi-sentence and a reference. Now the paragraph reads like:

While these strategies and tools are essential for global pandemic preparedness, there remains limitations and gaps in the current system, such as the use of convalescent blood blood products [30].”